# Does mandibular bone structure predict subsequent height loss? A longitudinal cohort study of women in Gothenburg, Sweden

Nivetha Natarajan Gavriilidou,[1] Grethe Jonasson [ID],[2] Valter Sundh,[3] Elisabet Rothenberg [ID],[4] Lauren Lissner[3]

[1]Department of Dental Medicine, Karolinska Institute, Stockholm, Sweden
[2]Department of Behavioral and Community Dentistry, University of Gothenburg, Gothenburg, Sweden
[3]School of Public Health and Community Medicine, Institute of Medicine, Sahlgrenska Academy, University of Gothenburg, Gothenburg, Sweden
[4]Department of Nursing and Integrated Health Sciences, Kristianstad University, Kristianstad, Sweden

**Correspondence to**
Dr Grethe Jonasson;
grethe.jonasson@gu.se

## ABSTRACT

**Background** Several risk factors for loss of height with increasing age have been identified.

**Objective** To investigate if mandibular bone structure predicts future height loss in middle-aged and elderly Swedish women.

**Design** Prospective cohort study with longitudinally measured heights, radiographical assessments of the cortical bone using Klemetti's Index (normal, moderate or severely eroded cortex) and classification of the trabecular bone using an index proposed by Lindh *et al* (sparse, mixed or dense trabeculation). No intervention was performed.

**Setting** Gothenburg, Sweden.

**Participants** A population-based sample of 937 Swedish women born in 1914, 1922 and 1930 was recruited. At the baseline examination, the ages were 38, 46 and 54 years. All had undergone a dental examination with panoramic radiographs of the mandible, and a general examination including height measurements on at least two occasions.

**Main outcome measure** Height loss was calculated over three periods 12–13 years (1968–1980, 1980–1992, 1992–2005).

**Main results** Mean annual height loss measures were 0.075 cm/year, 0.08 cm/year and 0.18 cm/year over the three observation intervals, corresponding to absolute decreases of 0.9 cm, 1.0 cm and 2.4 cm. Cortical erosion in 1968, 1980 and 1992 significantly predicted height loss 12 years later. Sparse trabeculation in 1968, 1980 and 1992 also predicted significant shrinkage over 12 or 13 years. Multivariable regression analyses adjusting for baseline covariates such as height, birth year, physical activity, smoking, body mass index and education yielded consistent findings except for cortical erosion 1968–1980.

**Conclusion** Mandibular bone structure characteristics such as severe cortical erosion and sparse trabeculation may serve as early risk factors for height loss. Since most individuals visit their dentist at least every 2 years and radiographs are taken, a collaboration between dentists and physicians may open opportunities for predicting future risk of height loss.

## INTRODUCTION

Height loss in older women is incrementally more pronounced at ages over 75 years. It has

## STRENGTHS AND LIMITATIONS OF THIS STUDY

⇒ The strengths are the prospective design, the population-based sample, the long follow-up, repeated measurements of height for the same individuals and that our findings can be applicable to many settings.

⇒ The main limitation is that loss of height could reflect different conditions, which may make accurate height estimates difficult.

⇒ The participation rates became lower in the later years, and the dropout for different reasons, mortality included, probably differs between age groups, which might impact the mean height losses.

been found to be associated with increased morbidity and mortality.[1–5] Several causes have been identified that lead to loss of height among older adults. Height loss could indicate age-associated progressive skeletal deformation such as kyphoscoliosis, flattening of plantar arch and altered posture,[6 7] and/or degenerative processes including osteoporosis, vertebral compression, attrition of intervertebral discs and vertebral fractures.[8] These alterations make accurate height estimation a challenge. Comorbidities of major height loss in older adults include functional impairments such as dependence in daily life activities, cognitive impairment and increased overall mortality risk.[1 5] Height loss over 5 cm has been shown to predict non-spinal fractures such as hip fractures and mortality independent of vertebral fractures.[1 6 7 9 10] Height loss in women exceeding 4 cm has been found to be related to fracture probability of over 50%.[3] In men, health consequences of height loss over 3 cm include increased risk of all-cause mortality and coronary heart disease.[9]

Osteoporosis is a widely acknowledged risk factor for future fractures in the elderly. According to the WHO, it is defined as bone density lower than 2.5 SDs under that of

young women using the bone mineral density of the hip with dual X-ray absorptiometry.[11] Simple, practical and relevant techniques are needed for early identification of the onset or extent of skeletal degeneration. Therefore, researchers have used oral variables such as optical density,[12] sparse trabeculation,[12–14] cortical width[15] and cortical erosion[15–17] for the prediction of osteoporosis and fracture.

Intraoral radiographs are usually an integral part of the routine oral examination. Assessment of mandibular structure, the cortical and trabecular bone pattern may be studied through visual inspection, hence providing a simple screening method for the evaluation of the bone condition. The hypothesis of the present study is that the mandibular bone contains sufficient information about the general bone condition to predict future height loss. The objective is to investigate two proxies for general bone mineral density, namely mandibular cortical erosion and trabecular coarseness, as predictors of height loss over subsequent periods of 12–13 years.

## METHODS
### Study population
The Prospective Population Study of Women in Gothenburg, Sweden is an ongoing longitudinal study initiated in 1968 in a cohort of women aged 38, 46, 50, 54 or 60 years. At that time, a representative sample of women in Gothenburg was identified using the Revenue Office Register with respect to the dates of birth. From the full baseline cohort, those born in 1914, 1922 and 1930 were included in the present analytical sample (flow chart, figure 1). The original survey included general health and dental examinations, which were repeated over the years. Each wave of survey was started in the fall and completed in the spring of the following year, that is, 1968–1969, 1980–1981, 1992–1993 and 2004–2005, and is hereafter referred to by its starting year. The participation rate in the initial examination was over 90%.[18 19] A non-participation analysis of the 24-year follow-up in 1992 showed that the women who declined to participate in the first survey did not differ significantly from participants except for the long-term survival, which was lower in the initial refusers. The sample remained representative of the general population in terms of parameters such as blood pressure, weight, height, body mass index (BMI), waist and hip circumference, serum cholesterol, triglyceride and urate.[19]

### Inclusion criteria
In total, 937 women born in 1914, 1922 or 1930 were included in the present analysis. The inclusion criteria were availability of at least one panoramic radiograph in 1968, 1980 or 1992 from which it was possible to assess the mandibular cortex erosion and sparsity of trabeculae, plus availability of a baseline and at least one subsequent

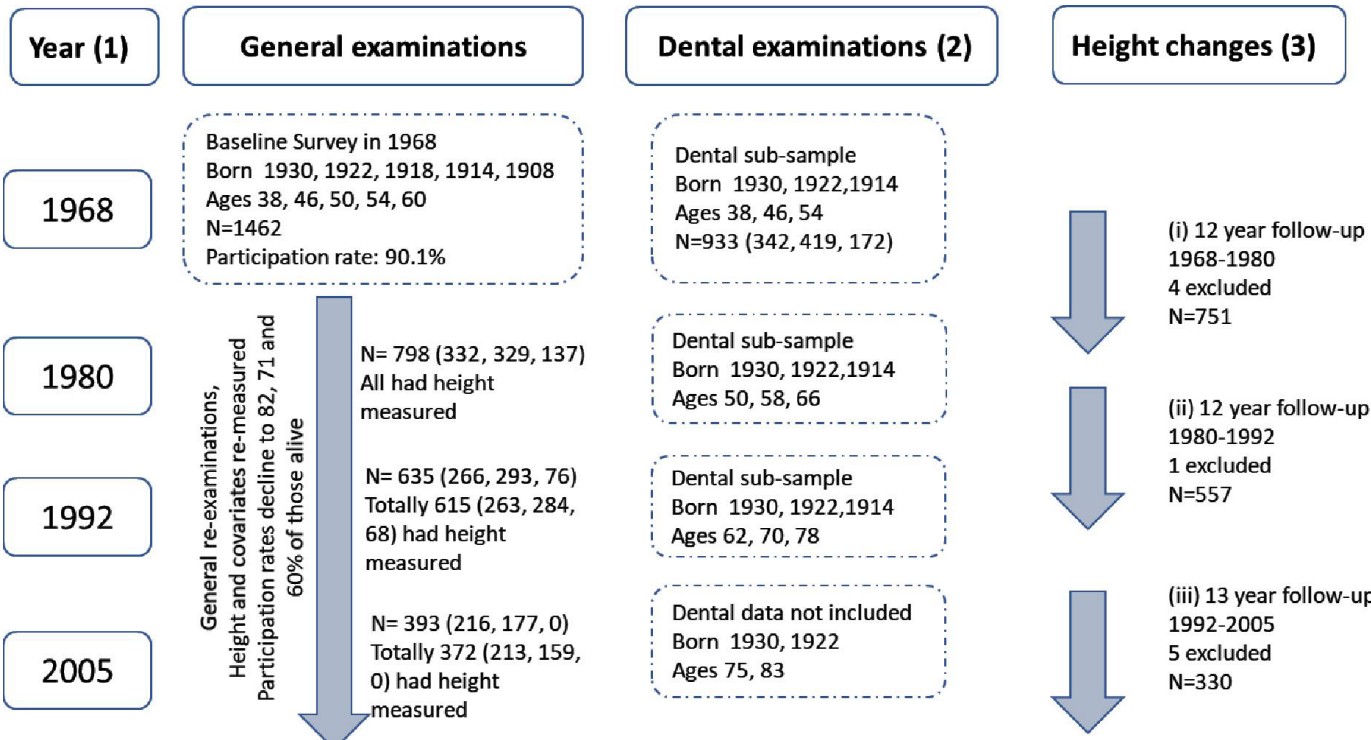

**Figure 1** Flow chart. (1) Each examination period starting in September and ending in June of the following year, that is, 1968–1969, 1980–1981, 1992–1993, 2005–2006. (2) Dental examination conducted shortly after general examination in three of original age groups. Mandibular bone structure (cortical and trabecular bone) is the main independent variable. (3) Heights were measured at general examinations, and the change in height was estimated at 12-year or 13-year intervals, after excluding any increases of over 2 cm.

height measure. In the first period (1968–1989), four individuals were excluded due to an *increase* in height of at least 2 cm. Number excluded in 1980–1992 was one, and it was five in the period 1992–2005. Results from previous observational studies using the same cohorts have shown that the number of participants should be sufficient for studying common health endpoints.

## Patient and public involvement
No patients involved.

## Main outcome
The main outcome variable was height loss during the 12–13 years following the dental examination. At all examinations, height and weight were assessed by standard anthropometric measurement techniques. A stadiometer was used to measure the participants' body heights (without shoes). Measurement of height was generally performed early in the day. Height loss was calculated as baseline height minus height at follow-up. Thus, a positive value indicated height loss.

Height change was assessed from follow-up examinations 12–13 years later, until 2005. The exclusion criterion was an observed *increase* in height of at least 2 cm during any of these follow-up periods, attributable to probable measurement errors.

There were three follow-up periods 12–13 years, namely 1968–1980, 1980–1992 and 1992–2005, where the first period comprised middle-aged women initially aged 38, 46 and 54 years, the second period with a subset of these women from ages 62, 70 and 78 years, and the third and the latest period starting at ages 75 and 83 years (no follow-up from 1914 birth cohort) (figure 1).

## Bone structure and other covariates
Radiographic assessment of mandibular bone structures was included as part of the dental examination. In the present analysis, the main independent variables were two components of mandibular bone status. On visual inspection of the panoramic radiograph of a mandible, the cortical bone was categorised using Klemetti's Index as mild/none (normal cortex), moderately or severely eroded cortex.[20] A normal cortex means an even endosteal margin; a moderately eroded cortex has semilunar defects in the endosteal margin, whereas a severely eroded cortex has heavy endosteal cortical porosities. The variable was dichotomised into 'severely eroded cortex' (1) and 'nonseverely eroded' (0). The second component, the trabeculation, was classified using an index proposed by Lindh *et al* as sparse, mixed or dense.[13 14] Sparse trabeculation means large intertrabecular spaces in the premolar area. Dense trabeculation has small intertrabecular spaces. Mixed dense and sparse trabeculation is mostly dense at the crest and sparse apically. The variable was dichotomised into 'sparse' trabeculation (1) and 'normal' (mixed and dense) trabeculation (0).

Baseline variables, such as height, BMI, smoking, physical activity and education, were considered potential covariates that could be associated with mandibular erosion and trabecular pattern, as well as height loss.[21–24] The survey questionnaire regarding smoking habit had six possible responses: never smoked (0), previous smoker but quit in the last 15 years (1), previous smoker but quit since last year (2), quit smoking since last year (3), current smoker without deep inhalation of the smoke (4) and current smoker with deep inhalation of the smoke (5). This variable was dichotomised to 'no smoking' (0) including response options 0, 1 or 2, and 'smoking' (1) including options 3, 4 or 5. The variable physical activity was categorised as (1) almost no physical activity or none (sedentary), (2) at least 4 hours/a week of physical activity, (3) regular physical activity and (4) regular intense training and competitions.[25] This variable was dichotomised as: 'physically inactive' (0) including the first response and 'physically active' (1) including the latter three choices. Education was considered a proxy for socioeconomic status as proposed by Kaufman and Cooper[26] and categorised as low: pre-high school (up to 9 years); medium: high school with ≤2 years at university; and high: 3 or more years at university. The variable was dichotomised into 'low education' (1) versus 'higher' (0).

## Statistical analyses
Mann-Whitney two-sample test was used for differences between two groups. In order to test the importance of lifestyle factors, linear multiple regression analyses were performed, where height loss was the dependent variable. The following variables were tested for independent prediction of height loss: the two dichotomous mandibular bone variables, smoking, physical activity, height, BMI, education and age (birth year). Height, height loss and BMI were continuous variables, whereas the two bone variables, smoking, physical activity and education were dichotomised as described above. The significance level was p<0.05. Statistical analyses were performed with SAS V.9.2 software (SAS Institute).

## RESULTS
Selected characteristics of the sample are shown in table 1. Mean age at the four height measurements (years 1968, 1980, 1992 and 2005) were 44.9, 56.9, 68.0 and 78.8 years, respectively, corresponding to three periods of height change (1968–1980, 1980–1992 and 1992–2005). There was a rising trend in the proportion of women with severe cortical erosion from 3.2% among the participants in the period 1968–1980, to 11.1% in the period 1980–1992 and up to 49.8% in the period 1992–2005. Similarly, prevalence of sparse trabeculation increased from 20.6% among the participants 1968–1980, to 33.5% in 1980–1992 and up to 41.6% in the period 1992–2005. At the same time, the mean height loss over the three

**Table 1** Descriptive characteristics of the population subsamples born in 1914, 1922 and 1930 at examination year 1968 followed up across periods 1968–1980, 1980–1992, 1992–2005

| | 1968 | 1980 | 1992 | 2005 |
|---|---|---|---|---|
| Age of women born 1930, years (N) | 38 (342) | 50 (289) | 62 (251) | 75 (190) |
| Born 1922 | 46 (419) | 58 (327) | 70 (239) | 83 (140) |
| Born 1914 | 54 (172) | 66 (135) | 78 (67) | 91 (0)* |
| | Total (933) | Total (751) | Total (557) | Total (330) |
| **Parameters at the end of the period** | **1968** | **1968–1980** | **1980–1992** | **1992–2005** |
| Mean age (SD) | 44.9 (5.8) | 56.9 (5.8) | 68.0 (5.4) | 78.8 (3.8) |
| Mean height in cm (SD) | 163.9 (6.0) | 163.0 (6.1) | 162.1 (6.2) | 160.4 (6.1) |
| Almost no physical activity, N (%) | 157 (16.8) | 221 (29.4) | 117 (21.0) | 25 (9.3) |
| Active smokers, N (%) | 308 (41.0) | 257 (34.3) | 115 (20.6) | 30 (9.2) |
| Education level† | | | | |
| Low | 630 (67.7) | 506 (67.4) | 364 (65.4) | 209 (63.3) |
| Medium | 251 (27.0) | 205 (27.3) | 159 (28.5) | 93 (28.2) |
| High | 49 (5.3) | 39 (5.2) | 33 (5.9) | 28 (8.5) |
| Mean body mass index (kg/m$^2$) | 23.7 (3.7) | 24.7 (3.9) | 26.4 (4.2) | 26.2 (4.5) |
| Weight (kg) | 63.6 (10.8) | 66.0 (11.2) | 69.0 (11.9) | 68.8 (11.8) |
| **Period 1968–1980** | | | | |
| Mean height loss (cm (SD)) | 0.90 (0.89) | | | |
| Mean annual height loss (cm/year) | 0.075 | | | |
| **Period 1980–1992** | | | | |
| Mean height loss (cm (SD)) | | 1.0 (1.21) | | |
| Mean annual height loss (cm/year) | | 0.08 | | |
| **Period 1992–2005** | | | | |
| Mean height loss (cm (SD)) | | | 2.4 (1.54) | |
| Mean annual height loss (cm/year) | | | 0.18 | |
| **Mandibular parameters at the start of the period** | | | | |
| Cortical erosion, N (%) | | | | |
| Mild/none | 583 (65.2) | 451 (62.9) | 175 (33.6) | 5 (8.2) |
| Moderate | 283 (31.7) | 243 (33.9) | 288 (55.3) | 128 (42.0) |
| Severely eroded | 28 (3.1) | 23 (3.2) | 58 (11.1) | 152 (49.8) |
| Trabeculation, N (%) | | | | |
| Dense | 236 (28.8) | 133 (20.3) | 78 (14.8) | 15 (9.3) |
| Mixed | 414 (50.6) | 302 (46.2) | 208 (39.5) | 79 (49.1) |
| Sparse | 169 (20.6) | 219 (33.5) | 241 (45.7)) | 67 (41.6) |

Cortical bone was classified using Klemetti's Index as mild, moderate or severe erosion, and the trabeculation classified into sparse, mixed or dense.
Number excluded based on shrinkage values over period 1968–1980: 4, 1980–1992: 1, 1968–1992: 4, 1992–2005: 5.
Not all participants with height measurements had an assessment of cortical erosion and even fewer had trabecular assessment due to edentulism.
*There were none in this age group in the follow-up period 1992–2005.
†Education level categories: low: pre-high school level (up to 9 years), medium: high school level with ≤2 years at university, high: 3 or more years at university.

survey periods was 0.9 cm, 1.0 cm and 2.4 cm, with corresponding annual decreases of 0.075, 0.08 and 0.18 cm/year (table 1).

Table 2 describes prospective analyses of association between mandibular bone structure at the three age periods and subsequent height loss, without covariate adjustment.

In all periods, height loss was highest in the groups with severe cortical erosion and the groups with sparse trabeculation (table 2). The difference in shrinkage between the group with normal versus severe erosion was 0.57 cm in the early period, 0.76 cm in the middle period and 0.46 cm in the last period (table 2). The difference in height loss between the group with normal/mixed and that with sparse trabeculation was 0.13 cm in the early period, 0.30 cm in the middle period and 0.56 cm in the last period. Multivariable linear regression analyses in table 3 confirmed the

**Table 2** Height loss in relation to cortical erosion and trabecular spacing

| Mandibular bone structure | Height loss (1968–1980) N=751 | | | Height loss (1980–1992) N=557 | | | Height loss (1992–2005) N=330 | | |
|---|---|---|---|---|---|---|---|---|---|
| **Cortical erosion** | **Mean±SD** | **Rate/year** | **P value** | **Mean±SD** | **Rate/year** | **P value** | **Mean±SD** | **Rate/year** | **P value** |
| Normal | 0.89±0.88 | 0.07 | 0.002 | 0.90±1.29 | 0.08 | <0.001 | 2.10±1.76 | 0.16 | 0.034 |
| Severe | 1.46±0.75 | 0.12 | | 1.66±1.15 | 0.14 | | 2.56±1.77 | 0.20 | |
| **Trabecular bone** | | | | | | | | | |
| Normal | 0.88±0.86 | 0.07 | 0.011 | 0.81±1.04 | 0.07 | 0.004 | 2.16±1.26 | 0.16 | 0.002 |
| Sparse | 1.01±0.99 | 0.08 | | 1.11±1.15 | 0.09 | | 2.72±1.82 | 0.20 | |

Cortical erosion was dichotomised into normal (mild or moderate) erosion and severe erosion. The trabeculation was classified into normal (mixed or dense) or sparse.

significant associations between the height loss and cortical erosion during the periods 1980–1992 and 1992–2005 but not 1968–1980, after adjusting for covariates as height, age (birth year), physical activity, smoking, education and BMI. The trabeculation showed a significant correlation with height loss in all three periods (table 3).

The prediction is based on cortical erosion and trabeculation (both dichotomous) adjusting for covariates height, year of birth, physical activity, smoking, education and BMI.

The importance of the covariates for predicting height loss is shown in table 3. Age is by far the most important parameter. Physical activity, baseline height and smoking showed inconsistent associations with height loss in different models, whereas BMI and education were not significant in any period.

## DISCUSSION
### Statement of principal findings
The present study was conducted among three systematically recruited birth cohorts from the Prospective Population Study of Women in Gothenburg, Sweden. We found that the longitudinal height loss was significantly predicted by mandibular bone parameters such as severe erosion of the cortex and sparse trabeculation. There was a clear increase in height loss over the follow-up time, in parallel with ageing and an increasing cortical erosion and sparseness of trabeculae.

**Table 3** Linear multivariable regression models predicting height loss over time intervals 1968–1980, 1980–1992 and 1992–2005

| | Height loss (1968–1980) N=751 | | Height loss (1980–1992) N=557 | | Height loss (1992–2005) N=330 | |
|---|---|---|---|---|---|---|
| | **Coefficient (SE)** | **P value** | **Coefficient (SE)** | **P value** | **Coefficient (SE)** | **P value** |
| Severe erosion | 0.20 (0.18) | 0.27 | 0.48 (0.19) | 0.011 | 0.43 (0.23) | 0.050 |
| Height at baseline | 0.01 (0.01) | 0.22 | 0.02 (0.01) | 0.051 | 0.03 (0.02) | 0.098 |
| Birth year | −0.05 (0.01) | <0.001 | −0.06 (0.01) | <0.001 | −0.02 (0.03) | 0.551 |
| Physical activity | −0.11 (0.06) | 0.07 | −0.13 (0.10) | 0.094 | −0.019 (0.17) | 0.907 |
| Smoking | −0.02 (0.01) | 0.27 | 0.06 (0.03) | 0.031 | 0.120 (0.07) | 0.072 |
| Education | 0.02 (0.07) | 0.81 | 0.15 (0.12) | 0.218 | 0.25 (0.23) | 0.273 |
| BMI | 0.01 (0.01) | 0.34 | 0.001 (0.02) | 0.929 | −0.015 (0.03) | 0.609 |
| Sparse trabeculation | 0.16 (0.08) | 0.038 | 0.24 (0.11) | 0.023 | 0.45 (0.22) | 0.040 |
| Height at baseline | 0.01 (0.01) | 0.056 | 0.02 (0.01) | 0.021 | 0.03 (0.02) | 0.131 |
| Birth year | −0.06 (0.01) | <0.001 | −0.06 (0.01) | <0.001 | −0.02 (0.03) | 0.450 |
| Physical activity | −0.14 (0.07) | 0.019 | −0.12 (0.07) | 0.106 | −0.10 (0.16) | 0.528 |
| Smoking | −0.02 (0.02) | 0.320 | 0.018 (0.02) | 0.439 | 0.13 (0.05) | 0.011 |
| Education | −0.02 (0.07) | 0.773 | 0.14 (0.11) | 0.190 | 0.20 (0.23) | 0.388 |
| BMI | 0.012 (0.01) | 0.239 | 0.02 (0.02) | 0.165 | −0.01 (0.03 | 0.717 |

Cortical erosion was dichotomised into severe erosion and normal (mild or moderate) erosion. The trabeculation was classified into sparse or normal (mixed or dense).
BMI, body mass index.

Our findings can be applicable to many settings. Swedish and Norwegian women have the highest frequencies of fragility fractures in the world[27]; that is, they probably have sparser trabeculation and more severe cortical erosion than women in other countries, but if for instance Japanese women have severely eroded cortex,[16] they may be of risk of height loss as well. Our main finding that severe cortical erosion and sparse trabeculation predict height loss builds on previous work in these cohorts showing mandibular bone structure to be a risk factor for future fractures.[24 28 29]

## Comparisons with other studies

Many previous studies have used height as a proxy for health to compare the welfare of populations, whereas few have investigated how not only height but also height loss could be associated with health in old age.[5 10 30 31] Height is considered a marker of early life health, whereas height loss is a potential risk factor for later life health.[5 10 30 31]

Huang *et al* estimated pre-shrinkage height and studied the association between height loss and different measures of health of older people.[10] They found that height loss for both men and women was negatively associated with better education, urban residence and household economy. Moreover, height loss was positively correlated with poor health outcomes, especially strong for cognition decline.[10] Mai *et al* found that age, use of oral corticosteroids and strenuous exercise were significantly associated with marked height loss.[30] Fernihough and McGovern found that height loss among older English men and women across a lifetime decreases by 2–4 cm on average.[31] These numbers are similar to our findings, where we found an average decrease of 4.3 cm after 37 years.

Fernihough and McGovern found that middle-aged and older Indonesian women exhibit considerable height loss despite being short to begin with, losing an average of 1.06 cm over 17 years.[5] Ethnicity, working in the agricultural sector and availability of local health infrastructure were correlated with loss of height.[5] Extreme height loss, greater than 3 cm, was associated with 8%–10% lower lung function and grip strength and with an increased mortality.[5] They also found that loss of height is higher among older age groups and taller individuals,[5] which is in accordance with our results. A recent investigation demonstrated, using national register data linked to the same cohort as described here, that loss of height over 12 years predicted mortality from all-cause and cardiovascular disease.[4] This is in line with the study of Jain and Ma.[5] Such data are not widely available in prospective epidemiological studies.

As described, height loss is associated with poor health.[1 4 5 8–10 30 31] Vertebral discs degenerate far earlier than other musculoskeletal tissues[1] and can lead to back pain and disability in the older adults[1] and in severe cases even vertebral fractures, which have been shown to have direct or indirect consequences such as decreased mobility, depression, hospitalisation and mortality.[22]

Panoramic dental radiography indices, cortical thickness, cortical erosion and sparse trabeculation have been suggested as screening tools for identifying risk groups with osteoporosis.[16 17 24] Cortical erosion, assessed at dental practices, has been shown to be useful in the identification of undetected low bone mineral density among postmenopausal women.[17] However, Okabe *et al* found no significant correlations between cortical erosion assessments and the occurrence of fractures in a group of 80-year-old Japanese men and women.[15] Swedish and Norwegian women have high frequencies of fragility fractures,[27] which may explain why Jonasson *et al* found significant associations between cortical erosion and fracture.[28] Another difference is that Okabe *et al* studied 5 years of fracture incidence, whereas Jonasson *et al* investigated the same over 10 years.[15]

Mandibular cortical width is associated with general bone mineral density,[16 32] but not with future osteoporotic fractures.[28] Sparse trabeculation was shown to be associated with a twofold higher risk of future fracture in a younger group and with a threefold to fourfold higher risk of future fracture in an older group.[29] As far as we know, relationships between height loss and oral indices have not been examined before.

All bone structures undergo an alteration in the bone remodelling mechanism in connection with menopause, leading to greater bone loss than bone formation.[33] The imbalance results in trabecular thinning, increase in intratrabecular spacing, loss of connectivity in the meshwork and intracortical erosion, thereby leading to diminished bone strength and increased risk of fracture.[34 35] Taller women have thinner cortical bone (in the appendicular skeleton) and therefore a greater risk of fractures.[36] The vertebral bone mass is largely trabecular with only a thin shell of cortical bone. The thin cortex together with the axial compressive load on the vertebra, distributed on fewer and thinner trabeculae, result in an increasing risk of fractures.[37 38]

## Strengths and limitations

The strengths are the prospective design, the population-based sample, the long follow-up and the potential applicability of our findings to many settings, since many individuals visit their dentist at least every 2 years and radiographs are taken. The unique aspect of this study is its design, consisting of long-term panel data on bone structure and height. The main limitations are the fact that height loss reflects various conditions that may make accurate height measures difficult. Moreover, participation rates became lower in the later years, and the dropout for different reasons, mortality included, probably differs between age groups, which might impact the mean height losses and the associations with covariates. The very low number of women with severely eroded cortex in the first period can explain why there was no significant association with height loss. This indicates that we cannot reach any reliable conclusion about the strength

of correlation between cortical erosion and height loss in the age interval 38–50.

## Implications of the findings

In our study population, the sparse trabeculation and the severe cortical erosion of the mandible could be seen to mimic the status of the vertebral trabeculae and intracortical erosion of the thin cortical shell of the vertebrae, which would, in turn, explain the potential height reduction. These two bone-related risk factors are important for the development of height loss as well as osteoporosis. A vertebral fracture is often asymptomatic, but it should not be ignored since it is a highly significant risk factor for future fractures. Therefore, using mandible status as a proxy and possible screening tool for the general status of vertebral trabeculae and intracortical bones is interesting for future health promotion.

However, implementation of new methods requires ethical considerations. Prevention has a long tradition in dentistry. Informing patients about their dental hygiene is an integral part of dental practice, whereas screening for osteoporosis/risk of height loss is not. An ethical dilemma in health promotion is that risk information may have unintended negative consequences for people who are currently free of symptoms. This is demonstrated in a qualitative study of Hvas et al,[39] whose interviews indicated that awareness of osteoporosis risk caused a feeling of worry in some postmenopausal women.

Visual assessments of the mandibular trabecular and cortical bone as conducted in this study are not routine in dental practice today, but simplified screening procedures using artificial intelligence may change this in the future.[40] Many postmenopausal women are interested in their health, and if they ask their dentist about the radiographs taken, prudence is needed when the bone is not normal due to sparse trabeculation or severely eroded cortex.

## CONCLUSION

Mandibular bone structure alterations such as severe cortical erosion and sparse trabeculation predicted height loss. They may therefore serve as proxy indicators when screening in the early phases of bone degenerative pathogenesis, signalling the ongoing bone remodelling and the need for further clinical attention to older women at risk of height loss.

**Contributors** NNG and LL initiated the study. VS and GJ did the acquisition of data. NNG, LL, VS, GJ and ER contributed to the analysis and interpretation of data. VS and NNG did the statistical analyses. NNG wrote the first draft of the manuscript. NNG, LL, GJ, ER and VS critically revised the manuscript and approved the final version. LL obtained the funding. NNG is responsible for the overall content as the guarantor and accepts full responsibility for the work and/or the conduct of the study, had access to the data, and controlled the decision to publish.

**Funding** This work was supported by grants from the Swedish Research Council for Health, Working Life and Welfare (Forte, EpiLife 2016-1506), and the Swedish ALF-agreement (ALFGBG 720201).

**Competing interests** None declared.

**Patient and public involvement** Patients and/or the public were not involved in the design, or conduct, or reporting, or dissemination plans of this research.

**Patient consent for publication** Not required.

**Ethics approval** This study involves human participants and was approved by the Regional Ethics Review Board of the University of Gothenburg (DNR 65-80, 179-92, T453-04). Participants gave informed consent to participate in the study before taking part.

**Provenance and peer review** Not commissioned; externally peer reviewed.

**Data availability statement** Data are available upon reasonable request.

**ORCID iDs**
Grethe Jonasson http://orcid.org/0000-0001-6770-0659
Elisabet Rothenberg http://orcid.org/0000-0002-3692-7014

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
