## [Reviewer comments · BMJ Open]

ARTICLE DETAILS

TITLE (PROVISIONAL)	Does mandibular bone structure predict subsequent height loss? A longitudinal cohort study of women in Gothenburg, Sweden
AUTHORS	Gavriilidou, Nivetha N; Jonasson, Grethe; Sundh, Valter; Rothenberg, Elisabet; Lissner, Lauren

VERSION 1 – REVIEW

REVIEWER	Cortés-Martín, Jonathan University of Granada
REVIEW RETURNED	12-Aug-2022

GENERAL COMMENTS	Congratulations for the work. It is a very interesting topic and provides interesting data to the scientific community. However, it needs some aspects of improvement. -The structure of the abstract needs to be organized. The abstract should contain the sections of background, objective, methodology, main result and conclusion to be complete.-The keywords need to be revised. They need to be more concrete and representative of the chosen topic.-The section on strengths and limitations of the study should be included after the discussion.-The statistical analysis performed needs to be described in more detail in the methodology section.-The discussion section should be revised. In this section it is necessary to compare and discuss the results obtained in the study with the rest of the literature published on the same subject, and this sometimes does not happen.- It would be of great interest to the reader of this article to propose possible future lines of research on this subject.
---

REVIEWER	Jain, Urvashi University of South Alabama Mitchell College of Business, Economics, Finance, and Real Estate
REVIEW RETURNED	21-Sep-2022

GENERAL COMMENTS	The paper examines whether two measures of mandibular bone structure are significant predictors of height loss among a representative sample of mid aged and older women in Sweden. The manuscript is well organized, with the use of long-term panel
---

data as its biggest strength, and the study's findings have a clear contribution. Listed below are a few comments which I hope the authors will find helpful.

Major comments

1. Analysis

Page 9, para 1: since one of the covariates is baseline height, using BMI (which uses height as a variable itself) as another covariates may not be ideal. The authors could instead replace BMI with a) weight or even better – change in weight over time, and/or b) grip strength since it helps capture muscle wasting (please see this paper for more details: “Physical stature decline and the health status of the elderly population in England” <https://doi.org/10.1016/j.ehb.2013.12.010>).

2. Results

a. Table 1: what do the two asterisks next to cortical erosion and trabeculation indicate? These are not elaborated upon in the table footnotes.

b. Table 2: the authors could consider mentioning the p-values in the table, along with using the *, **, *** notation to indicate statistical significance at 10%, 5% and 1% levels respectively, instead of using footnotes c), d), and e) and listing p-values in the footnotes.

c. Table 3:

Since cortical erosion was shown to have three categories in tables 1 and 2 – mild/none, moderate, and severe, what does “cortical erosion” represent in Table 3? Moderate and severe combined? This should be clarified in the table footnotes, and the reference/omitted group mentioned.

Education was also shown to have three groups, but only one coefficient is reported in Table 3. Please clarify/correct this. This also made me wonder how the authors reached the conclusion that “education was a risk factor” on page 13, since we don't know what group of education we're looking at in Table 3.

To maintain consistency in presentation of results, since the *** notation to indicate $p < 0.001$ was used in Table 2, the authors could also use it in Table 3.

Last, I wasn't sure why one of the coefficients on “sparse trabec.” was not bold font when all other mandibular bone coefficients were, even the insignificant ones.

3. Missing literature:

Page 5, para 1, last sentence – since this mentions prior studies on the health and mortality consequences of height loss, please see this paper, “Height shrinkage, health and mortality among older adults: Evidence from Indonesia”

<https://doi.org/10.1016/j.ehb.2020.100863>, which also found height loss above 3cm to be a predictor of all-cause mortality among a sample of mid aged and older women in Indonesia. This paper would also be an appropriate reference on page 14, para 1, when discussing previous findings on height loss and mortality.

	Page 14, para 3: “Height shrinkage is associated with poor health” the following references ought to be added here: “Height shrinkage, health and mortality among older adults: Evidence from Indonesia” https://doi.org/10.1016/j.ehb.2020.100863 “Health, Height, Height Shrinkage, and SES at Older Ages: Evidence from China” https://www.aeaweb.org/articles?id=10.1257/app.5.2.86 4. Study sample size: page 6, last para, inclusion criteria mentions 1003 women but figure 1 mentions the dental sub-sample as 933. Please clarify which sample is the 1003 women. 5. Language: a. Page 6, para 1: please provide full form of BMD before switching to abbreviation. b. Page 9, para 1: when listing covariates, please consider using “mandibular bone variables” instead of just “bone variables” to be more precise. c. Page 12, last para: “severely cortical erosion” should be “severe cortical erosion”. d. Page 13, para 2: “Cortical erosion was the best risk factor” may not be an appropriate way to present this finding. “Best” could be replaced by “better”. e. Page 14, para 3: the authors state that “More years of education predicted more stature shrinkage in the latest observation period only, which may be related to societal trends towards longer educations for women.” I am not sure I agree. While there is indeed an increasing trend in educational attainment among women over time, how does that explain why higher education may be related to greater shrinkage? Height shrinkage in fact is highest among oldest women, who are more likely to be less educated. I urge the authors to either explain this statement more clearly, or drop this explanation.
--	---

VERSION 1 – AUTHOR RESPONSE

Reviewer 1 wrote: The structure of the abstract needs to be organized. The abstract should contain the sections of background, objective, methodology, main result and conclusion to be complete.

Our answer: We have checked the latest issues of BMJ Open for the format and the abstract is completed with “Background”. Page 3

Reviewer 1 wrote: The keywords need to be revised. They need to be more concrete and representative of the chosen topic.

Our answer: the keywords are revised, and the new keywords are: Ageing, body height, cortical bone, dental radiography, mandible, stature decline, trabecular bone, women’s health. Page 4.

Reviewer 1 wrote: The section on strengths and limitations of the study should be included after the discussion.

Our answer: We have reorganised the discussion. In the author instructions on the website, it is written: “Please include a ‘Strengths and limitations of this study’ section after the abstract.”

Reviewer 1 wrote: The statistical analysis performed needs to be described in more detail in the methodology section.

Our answer: we have now added more details about the variables. Page 10.

Reviewer 1 wrote: The discussion section should be revised. In this section it is necessary to compare and discuss the results obtained in the study with the rest of the literature published on the same subject, and this sometimes does not happen.

Our answer: The discussion is revised, and two new references added. Some parts have been moved and especially the part with comparisons with the rest of the literature is increased considerably.

Reviewer 1 wrote: It would be of great interest to the reader of this article to propose possible future lines of research on this subject.

Our answer: Just before the conclusion, we have added:” Therefore, these results highlight the need for implementation studies in in which dental practitioners can become active participants in identifying individuals at risk of shrinkage, osteoporosis, and fractures, in collaboration with other clinicians.”

Reviewer 2 wrote

1. Analysis

Page 9, para 1: since one of the covariates is baseline height, using BMI (which uses height as a variable itself) as another covariates may not be ideal. The authors could instead replace BMI with a) weight or even better – change in weight over time, and/or b) grip strength since it helps capture muscle wasting (please see this paper for more details: “Physical stature decline and the health status of the elderly population in England” <https://doi.org/10.1016/j.ehb.2013.12.010>).

Our Answer: We see your point but we do not have data on grip strength. Weight is not related to height loss in our dataset; but it is added in table 1 now. BMI is frequently used in fracture papers, and we employ it as a measure of underweight/normal weight/overweight. Fractures (including vertebral fractures) are often seen in women with low BMI, but less often in women with high BMI; they have fracture of the lower bone more often.

2. Results

a. Table 1: what do the two asterisks next to cortical erosion and trabeculation indicate? These are not elaborated upon in the table footnotes.

Our answer: Sorry, they should not be there and are deleted now.

b. Table 2: the authors could consider mentioning the p-values in the table, along with using the *, **, *** notation to indicate statistical significance at 10%, 5% and 1% levels respectively, instead of using footnotes c), d), and e) and listing p-values in the footnotes.

*Our answer: We have now included the p-values in the tables instead of *, ** or ***.*

c. Table 3:

Since cortical erosion was shown to have three categories in tables 1 and 2 – mild/none, moderate, and severe, what does “cortical erosion” represent in Table 3? Moderate and severe combined? This should be clarified in the table footnotes, and the reference/omitted group mentioned.

Our answer: Cortical erosion in table 3 represents all three categories. No erosion group is omitted. This is stated in the foot note: Mandibular cortical bone was categorized using Klemetti’s Index as normal, moderate, or severe eroded cortex and the trabeculation classified into sparse or normal (mixed or dense. By mistake we wrote sparse trabeculation; it was wrong, it should be just trabeculation, and this is the cause of the confusion concerning erosion. We kept all three categories in the multiple analysis.

Education was also shown to have three groups, but only one coefficient is reported in Table 3. Please clarify/correct this. This also made me wonder how the authors reached the conclusion that “education was a risk factor” on page 13, since we don’t know what group of education we’re looking at in Table 3.

Our answer: Yes, you are right. Because we wrote sparse trabeculation there was a confusion both concerning erosion and education, but also here, all three education categories are included in the multiple regression analysis. The conclusion concerning high education as a risk factor was drawn

using Table 2. However, we have dropped our explanation in the text. (Probably our findings are explained by the old data: the cohorts born 1914, 1922 and 1930 are likely to have another lifestyle than women born 30-50 years later).

To maintain consistency in presentation of results, since the *** notation to indicate $p < 0.001$ was used in Table 2, the authors could also use it in Table 3.

Our answer: now we have done so here as well.

Last, I wasn't sure why one of the coefficients on "sparse trabec." was not bold font when all other mandibular bone coefficients were, even the insignificant ones.

Our answer: now all are bold.

3. Missing literature:

Page 5, para 1, last sentence – since this mentions prior studies on the health and mortality consequences of height loss, please see this paper, "Height shrinkage, health and mortality among older adults: Evidence from Indonesia" <https://doi.org/10.1016/j.ehb.2020.100863>, which also found height loss above 3cm to be a predictor of all-cause mortality among a sample of mid aged and older women in Indonesia. This paper would also be an appropriate reference on page 14, para 1, when discussing previous findings on height loss and mortality.

Our answer: Thank you very much for the useful references. Now we have added the references of Fernihough A, McGovern ME and Jain U & Ma M. They are discussed together with the reference Huang et al. (our ref 9).

Page 14, para 3: "Height shrinkage is associated with poor health" the following references ought to be added here: "Height shrinkage, health and mortality among older adults: Evidence from Indonesia" <https://doi.org/10.1016/j.ehb.2020.100863>, and "Health, Height, Height Shrinkage, and SES at Older Ages: Evidence from China" <https://www.aeaweb.org/articles?id=10.1257/app.5.2.86>

Our answer: yes now these references are discussed.

4. Study sample size: page 6, last para, inclusion criteria mentions 1003 women but figure 1 mentions the dental sub-sample as 933. Please clarify which sample is the 1003 women.

Our answer: The Gothenburg study is a bit complicated. 1003 had the bone structure assessments and height measurements but not all participated 1968. (New participants were included 1980 and some participants declined 1980). For simplicity, the number 937 are written in the text now, meaning that in 1968 we had 937 but four were excluded, giving the 933 in Table 1.

5. Language:

a. Page 6, para 1: please provide full form of BMD before switching to abbreviation. *Done.*

b. Page 9, para 1: when listing covariates, please consider using "mandibular bone variables" instead of just "bone variables" to be more precise. *Done.*

c. Page 12, last para: "severely cortical erosion" should be "severe cortical erosion". *Corrected*

d. Page 13, para 2: "Cortical erosion was the best risk factor" may not be an appropriate way to present this finding. "Best" could be replaced by "better". *Corrected*

e. Page 14, para 3: the authors state that "More years of education predicted more stature shrinkage in the latest observation period only, which may be related to societal trends towards longer educations for women." I am not sure I agree. While there is indeed an increasing trend in educational attainment among women over time, how does that explain why higher education may be related to greater shrinkage? Height shrinkage in fact is highest among oldest women, who are more likely to be less educated. I urge the authors to either explain this statement more clearly, or drop this explanation. *We have dropped the explanation.*

VERSION 2 – REVIEW

REVIEWER	Cortés-Martín, Jonathan University of Granada
REVIEW RETURNED	28-Nov-2022

GENERAL COMMENTS	First of all I would like to congratulate the authors for this work. A report on an interesting and attractive topic that provides interesting data to the scientific community. The title is appropriate, the abstract is organised and the keywords are concrete and related to the topic. The introduction is acceptable, perhaps a little brief, but orderly and justified. In the methodology section, it would be interesting to include a section on the ethical aspects of the study. The results are shown extensively and in detail in tables and figures. The discussion section is very interesting, but at times it is difficult to read, perhaps due to the length of the section. Some aspects need to be synthesised. The conclusion is correct. To make this report more attractive, I would add a section on the applicability of the study at present in your own field of work. The bibliographic style needs to be revised. It can be seen that the article has improved a lot with the revisions proposed by the previous reviewers. Good work.
---

REVIEWER	Jain, Urvashi University of South Alabama Mitchell College of Business, Economics, Finance, and Real Estate
REVIEW RETURNED	26-Nov-2022

GENERAL COMMENTS	I thank the authors for their revisions. The revised discussion section looks particularly improved. However, two comments from my initial report were not properly addressed in the latest version of the manuscript. Please see my comments below. 1. Properly reporting categorical coefficients: In Comment 2c (“Results”) of my initial report, I had asked about coefficients on cortical erosion, and on education. In their response, the authors state that “Cortical erosion in table 3 represents all three categories. No erosion group is omitted” – how can one coefficient on Cortical erosion represent three categories? This is possible if this variable was used as a continuous measure in the analysis, which would be incorrect. The authors in their response also stated that “all three education categories are included in the multiple regression analysis” – if that is the case, these variables must be used as categorical/dummy variables and Table 3 should show the following:
--

	1.1. Separate coefficients on moderate erosion and severe erosion, in comparison to mild/no erosion. 1.2. Sparse trabeculation should be listed in the table instead of just trabeculation, mentioning that normal trabeculation is the reference/omitted category in table notes. 1.3. Separate coefficients on medium and high education, in comparison to low education level, mentioning in tables notes that low education level is the reference/omitted category. In the current version, there is a single coefficient on education, which would have made sense if education was a contiguous measure like years of education, but that's not the case. 2. Appropriate citation within the draft: While it is good to see that the authors included the recommended prior research in references in the revised draft, these ought to be incorporated more fittingly in the text. In comment 3 ("missing literature") of my initial report, I had recommended that the authors cite the paper "Height shrinkage, health and mortality among older adults: Evidence from Indonesia" in the 1st paragraph of the introduction. I recommended this since a) the second sentence mentions that height loss is associated with increased morbidity and mortality and b) the last sentence cites a paper which examined height loss and mortality among men. Given that the submitted manuscript specifically looks at height loss among older women, the Indonesia paper should be cited in this paragraph since one of the main findings of that paper was that extreme height loss is associated with higher mortality among older women. 3. Language. Abstract, Main results: should be "covariates such as height". The manuscript would also benefit from a careful round of proof-reading/copyedit before it is suitable for publication.
--	--

VERSION 2 – AUTHOR RESPONSE

Point to point 2023

Reviewer 2 wrote:

1. Properly reporting categorical coefficients:

In Comment 2c ("Results") of my initial report, I had asked about coefficients on cortical erosion, and on education. In their response, the authors state that "Cortical erosion in table 3 represents all three categories. No erosion group is omitted" – how can one coefficient on Cortical erosion represent three categories? This is possible if this variable was used as a continuous measure in the analysis, which would be incorrect. The authors in their response also stated that "all three education categories are included in the multiple regression analysis" – if that is the case, these variables must be used as categorical/dummy variables and Table 3 should show the following::

- 1.1. Separate coefficients on moderate erosion and severe erosion, in comparison to mild/no erosion.
- 1.2. Sparse trabeculation should be listed in the table instead of just trabeculation, mentioning that normal trabeculation is the reference/omitted category in table notes.
- 1.3. Separate coefficients on medium and high education, in comparison to low education level, mentioning in tables notes that low education level is the reference/omitted category. In the current version, there is a single coefficient on education, which would have made sense if education was a contiguous measure like years of education, but that's not the case.

Our answer:

In Table 3, we have presented results of multivariable linear regression models. Our goal here is to adjust for the key covariates and investigate if the findings on cortical or trabecular bone are attenuated by any of these confounders. In the medical tradition, linear regression analyses can include both continuous and categorical data as independent variables. Sex (two categories), blood group (four categories), and age (continuous variable) can be considered in the same model. We acknowledge that in econometric modelling, different traditions may apply.

Now Table 2 is changed. In Table 2, we present height loss in relation to three categories of cortical erosion and two categories of trabecular spacing. P-values from Mann-Whitney test (MW) and Test of trend over three levels (Lin). The three-level scale used for cortical erosion will produce an effect measure that is an averaged effect of the difference (in the outcome variable) between category 1 and 2 and between category 2 and 3, thereby testing the null hypothesis. This has been clarified in the methods section.

The findings were unchanged after dichotomising education as mandatory vs higher attainment, (data not shown).

Reviewer 2 wrote:

2. Appropriate citation within the draft:

While it is good to see that the authors included the recommended prior research in references in the revised draft, these ought to be incorporated more fittingly in the text. In comment 3 (“missing literature”) of my initial report, I had recommended that the authors cite the paper “Height shrinkage, health and mortality among older adults: Evidence from Indonesia” in the 1st paragraph of the introduction. I recommended this since a) the second sentence mentions that height loss is associated with increased morbidity and mortality and b) the last sentence cites a paper which examined height loss and mortality among men. Given that the submitted manuscript specifically looks at height loss among older women, the Indonesia paper should be cited in this paragraph since one of the main findings of that paper was that extreme height loss is associated with higher mortality among older women.

Our answer: yes, this is much better. Now the Indonesian paper is reference number 5.

The reviewer wrote:

3. Language. Abstract, Main results: should be “covariates such as height”. The manuscript would also benefit from a careful round of proof-reading/copyedit before it is suitable for publication.

Our answer: now “such as” is added under main results and the manuscript is proof-read again.

Reviewer 1 wrote:

In the methodology section, it would be interesting to include a section on the ethical aspects of the study.

Our answer:

After the manuscript there is added information about ethical approval. Since we feel that the most interesting ethical aspects of the study concerns how it can be used in the clinic, we have chosen to write about this aspect under Discussion. We are of course aware that it is way too early to implement our results. It is not up to dentists to diagnose osteoporosis and risk for future fracture and height loss, but with prudence we may answer questions about radiographic traits.

Reviewer 1 wrote:

The discussion section is very interesting, but at times it is difficult to read, perhaps due to the length of the section. Some aspects need to be synthesised.

Our answer:

We have now simplified a bit by omitting previous references 37 and 38 about osteogenic potential of mandibular vs. long-bone marrow stromal cells, and osteoclastogenic potential. Furthermore, we have tried to shorten down and synthesise how we imagine that the degenerative processes in the spine lead to height loss.

Reviewer 1 wrote:

To make this report more attractive, I would add a section on the applicability of the study at present in your own field of work.

Our answer

Now we have added a section with some ethical aspects regarding the applicability of the study in the dental clinic.

Reviewer 1 wrote:

The bibliographic style needs to be revised.

Our answer:

Now we have changed the names of the journals to non-italic.

VERSION 3 – REVIEW

REVIEWER	Jain, Urvashi University of South Alabama Mitchell College of Business, Economics, Finance, and Real Estate
REVIEW RETURNED	08-Mar-2023

GENERAL COMMENTS	Major comment: 1. It is unfortunate and inefficient that the authors are yet to satisfactorily address my comment from round 1 on differentiating between continuous versus categorical variables when it comes to reporting and interpreting regression coefficients for these two types of variables in multiple regression analysis. The authors added a justification in the methods section, and state in their response letter that “In the medical tradition, linear regression analyses can include both continuous and categorical data as independent variables. Sex (two categories), blood group (four categories), and age (continuous variable) can be considered in the same model. We acknowledge that in econometric modelling, different traditions may apply.” My response is that it is common practice across fields that multiple linear regression analyses can include both continuous
--

	and categorical variables – using both kinds of variables is not an issue, and not what I have been saying. It is not about “different traditions” in econometrics either. All I am simply saying is: please be consistent and use variables you define as categorical in the section on “Bone structure and other covariates” as categorical in regression analyses as well (i.e., use dummy variables). This lack of consistency is apparent when the reader sees summary stats for a categorical variable in Table 2, which is subsequently treated as a continuous variable in Table 3. A bigger concern is, while using a variable as continuous or dummy in regression analysis does not matter if the variable is dichotomous, it does matter when there are three or more categories (as is the case for education and cortical erosion). Hence, I request the authors to satisfactorily address comments 1.1 and 1.3 from my previous report (i.e., use dummy variables in regression analysis and report those coefficients for categorical measures) and update the text in the results section accordingly. 1a. I wasn't sure why the authors mention “Sex (two categories), blood group (four categories),” in their response – the sample consists only of women, and blood group is not mentioned in the manuscript at all! Minor comment: 1. 1st, 3rd, and 4th line of section Statistical analyses should be moved to the prior section on Bone structure.
--	--

VERSION 3 – AUTHOR RESPONSE

Thank you for the possibility to improve the manuscript. New text is marked in yellow. The text marked in red is supposed to be deleted.

1. Reviewer 2 asked us to “be consistent and use variables you define as categorical in the section on “Bone structure and other covariates” as categorical in regression analyses as well (i.e., use dummy variables). This lack of consistency is apparent when the reader sees summary stats for a categorical variable in Table 2, which is subsequently treated as a continuous variable in Table 3. A bigger concern is, while using a variable as continuous or dummy in regression analysis does not matter if the variable is dichotomous, it does matter when there are three or more categories (as is the case for education and cortical erosion).”

Our answer: Yes, now we have tried being more consistent. Therefore, in the descriptive Table 1, we have presented values for height loss for all three trabeculation categories using the categorisation method of Lindh et al., and we used the 3-level categorisation proposed by Klemetti for the cortical erosion. Thereafter, the cortical erosion and trabeculation are consistently dichotomised in both Table 2 and Table 3 as explained in the revised text and footnotes. Similarly, education is presented with the three categories in Table 1 and dichotomised in Table 3.

2. The reviewer wrote. “Hence, I request the authors to satisfactorily address comments 1.1 and 1.3 from my previous report (i.e., use dummy variables in regression analysis and report those coefficients for categorical measures) and update the text in the results section accordingly.”

Our answer: Now we have dummy variables for cortical erosion, trabecular sparseness, and education. The coefficients are therefore changed, and the text changed accordingly.

3. The reviewer wrote “1a. I wasn’t sure why the authors mention “Sex (two categories), blood group (four categories),” in their response – the sample consists only of women, and blood group is not mentioned in the manuscript at all!”

Our answer: The mention was thought to exemplify.

4. Minor comment:

1. 1st, 3rd, and 4th line of section Statistical analyses should be moved to the prior section on Bone structure.

Our answer: Now the Statistical analyses section and the Bone structure section are changed in accordance with the Reviewer’s suggestion.

VERSION 4 – REVIEW

REVIEWER	Jain, Urvashi University of South Alabama Mitchell College of Business, Economics, Finance, and Real Estate
REVIEW RETURNED	04-May-2023
GENERAL COMMENTS	Thank you for addressing my comments and revising the manuscript.